# Clean Production of Biofuel from Waste Cooking Oil to Reduce Emissions, Fuel Cost, and Respiratory Disease Hospitalizations

**José Carlos Curvelo Santana** [1,2,*], **Amanda Carvalho Miranda** [1], **Luane Souza** [2], **Charles Lincoln Kenji Yamamura** [1], **Diego de Freitas Coelho** [3], **Elias Basile Tambourgi** [3], **Fernando Tobal Berssaneti** [1] and **Linda Lee Ho** [1]

1 Department of Production Engineering, University of São Paulo, São Paulo 05508-010, SP, Brazil; mirandaca1@hotmail.com (A.C.M.); charles.yamamura@usp.br (C.L.K.Y.); fernando.berssaneti@usp.br (F.T.B.); lindalee@usp.br (L.L.H.)
2 Industrial Engineering Post Graduation Program, Federal University of ABC, São Bernardo do Campo, São Paulo 09606-045, SP, Brazil; luane.souza@aluno.ufabc.edu.br
3 School of Chemical Engineering, State University of Campinas, São Paulo 13083-970, SP, Brazil; diegofcoelho@gmail.com (D.d.F.C.); eliastam@feq.unicamp.br (E.B.T.)
* Correspondence: jose.curvelo@ufabc.edu.br

**Abstract:** Renewable energies are cleaner forms of energy, and their use, has intensified in recent decades. Thus, this work presents a proposal for reducing the emissions, fuel cost, and respiratory disease hospitalizations using environmental cost accounting principles to produce biodiesel production from waste frying oil. In our methodology, we conducted surveys, and collected waste cooking oil samples from local households and restaurants in São Paulo city, Brazil. Then, we produced biodiesel using these samples. Data on air pollutants were collected and correlated with the number of hospitalizations for respiratory diseases and their costs. Our results indicate that 330,000 respiratory disease hospitalizations were recorded in São Paulo city between 2009 and 2018, and the total cost for the Brazilian government reached US $117 million. Improving the city air quality by switching from fossil fuels to biodiesel could reduce the annual number of hospitalizations to 9880 and cost US $3.518 million, because the amount of pollutants emitted from burning fossil fuels was positively correlated with the number of respiratory disease hospitalizations and their costs. Moreover, the emission rates of particulate matter with particles less than 10 and 2.5 μm in diameter exceeded the World Health Organization limits throughout the study period. Using the survey data, we estimated that the average monthly quantity of waste cooking oil was 9794.6 m$^3$, which could generate 9191.2 m$^3$ of biodiesel and produce 239,713 t $CO_2$ of carbon credits. Environmental cost accounting revealed that it would be possible to achieve an annual profit of approximately US $300 million from the sale of excess biodiesel, carbon credits, and glycerine, and fuel acquisition savings which could improve the image of São Paulo city and quality of life of its residents. Thus, we present this as a way to reduce cost and hospitalizations, and increase the number of available hospital beds for other diseases, such as COVID-19.

**Keywords:** environmental cost accounting; human health care; greenhouse gas emission; biodiesel; waste cooking oil; air pollution

## 1. Introduction

### 1.1. Advantages and Disadvantages of Diesel Oil

A clean technology practice is presented in this work, which contributes to sustainable development by using a renewable fuel that is a cleaner technology than fossil fuels, as it reduces the number of polluting gases in the atmosphere, improving the health and quality of life of citizens. Among the main barriers, the proof of the economic or environmental feasibility of the proposals is cited, which is easily solved using the theory of environmental cost accounting [1–3]. The use of renewable energy sources is one of the cleanest ways the energy sector uses to contribute to sustainable development. Thus, the switch from

the source of fossil fuels (more pollutants) to renewable sources (e.g., biodiesel) has been intensified in recent decades [4–7].

Diesel engines, which are the most popular engines worldwide, are used in trucks, buses, trains, ships, and industrial off-road vehicles, as well as power generators and industrial machinery and equipment [8,9]. Their low operating costs, energy efficiency, high durability, and reliability have contributed to their high usability; however, diesel engines also present disadvantages. Diesel exhaust is a complex mixture of hydrocarbons, gases, sulfur, and particulates, that is produced during the combustion of diesel oil; moreover, diesel exhaust is one of the major contributors to the total aerosol mass and particulate matter (PM) in urban agglomerations, and is associated with high-traffic areas [7,9,10]. Although less than 1% of the exhaust gases from diesel engines are pollutants, they are responsible for severe environmental and health problems [8]. Thus, expanding the use of diesel engines has been contributing to the increase in atmospheric pollutant emissions and the worsening air quality and human health [9]. In addition, Hajjari et al. [11] stated that the gases emitted from burning diesel oil were responsible for 75.1%, 63.7%, 43%, 36.3%, and 1.8% of the atmospheric emissions of PM, nitrogen dioxide ($NO_2$), nitrogen oxides (NOx), sulfur dioxide ($SO_2$), and carbon monoxide (CO), respectively.

Selley et al. [12] indicated that the combustion of biodiesel instead of diesel oil could be used to reduce the diesel engine-related toxic emissions, because the oxygen content of biodiesel is higher than that of diesel oil; moreover, the combustion of biodiesel is complete, it does not generate aromatic or sulfurous compounds, and the amounts of unburned hydrocarbons, polycyclic aromatic hydrocarbons, CO, and PM generated by burning biodiesel are 90%, 75–90%, 43%, and 55%, respectively, lower than those generated by burning diesel oil.

Da Silva et al. [13], Reis et al. [14], and Silva Filho et al. [15] used biodiesel in power generators and noted that there were no significant differences between the amounts of power generated using the same amount of biodiesel blends and diesel oil. In addition, the greenhouse gas emissions associated with biodiesel were at least 30% lower than those associated with diesel oil. Biodiesel has also been tested in agricultural tractor engines, and the results indicated that engine efficiency achieved when using blends of up to 50% biodiesel was similar to that achieved when using diesel oil [13,14].

Hasan and Rhaman [16] demonstrated that the use of different percentages of biodiesel produced from waste oils shortened the ignition time of combustion engines, which promoted the reaction time of the mixtures and eventually reduced the emission level of unburned hydrocarbons. Furthermore, Hansan and Rhaman [17] indicated that compared to diesel oil, biodiesel obtained from waste oils provided shorter ignition delays, lower heat release rates, and slightly higher efficiency; in addition, to obtain a reduction in hydrocarbons, CO, and PM emissions. Pereira et al. [9] reported that the biggest benefit of using biodiesel was the significantly lower emission of pollutants, such as PM, CO, hydrocarbons, and NOx, compared to those generated by diesel oil.

Silva Filho et al. [15] and Chua et al. [18] demonstrated that using waste cooking oil to produce biodiesel was environmentally beneficial, and the $SO_2$, NOx, $N_2O$, $CO_2$, PM, volatile organic compounds (VOCs), and methane ($CH_4$) emissions generated by burning biodiesel were 100%, 97.95%, 96.08%, 95.42%, 99.99%, 91.52%, and 82.28%, respectively, lower than those generated by burning diesel oil. This corroborated the importance of using waste cooking oil to produce biodiesel, because biodiesel is renewable, more sustainable than its source, and more environmentally friendly than fossil fuel. Moreover, as waste cooking oil is a waste, it does not compete with edible oil in elevation and consumption [15], does not require an increase in soybean acreage, and does not contribute to the need for deforestation to increase acreage [15,19].

Santos et al. [20] indicated that Brazilian biodiesel consisted exclusively of soybean biodiesel and argued the need for diversifying its production to involve other oil sources, and thus, as reported by Živković et al. [21], the social benefits of biodiesel production would extend to the poor Brazilian communities [22,23].

Brodny and Tutak [24] indicated that gaseous pollutants are one of the most dangerous types of pollutants for human health, because of their lasting negative effects on human health and life expectancy. In addition, more than 7 million annual premature deaths are associated with gaseous pollutants, and therefore, they represent one of the biggest risks to human health worldwide [10].

Furthermore, De Marco et al. [25] reported that in 2005, particulate matter with particles less than 10 μm in diameter ($PM_{10}$) and $NO_2$ induced approximately 58,000 cardiovascular— and respiratory disease-related premature deaths in Italy. In China, more than one million annual premature deaths are attributed to air pollution, the pollution-related disease records increased by 33%, and the number of disabled individuals reached 76 million and continues to increase [26]. Furthermore, Yu et al. [26] indicated that Northern, Central, and Southern China are the regions where human health is more severely affected owing to PM, particulate matter with particles less than 2.5 μm in diameter ($PM_{2.5}$), $NO_2$, and $SO_2$ pollution, which is attributed to the large number of vehicles and human activities in these highly populated regions.

According to the World Health Organization [10], improving air quality by replacing fossil fuel with renewable fuels, such as biodiesel, could reduce pollutant emissions and respiratory disease hospitalizations by 30%; moreover, respiratory disease-associated mortality could also decrease significantly.

According to the Sistema Único de Saúde (SUS) database, air pollution cost in Brazil between 1993 and 1995 exceeded USD 2 million [27]. The money was spent on treating patients who developed diseases directly related to the excess pollutants. This amount of money could have covered the cost of 784,000 medical consultations or 10,100 births in Brazilian Health System (SUS)-affiliated hospitals. Research has reported the effects of gaseous emissions on human health in Brazil, as well as correlating the number of pollutant-related respiratory disease hospitalizations with their costs for the public health system SUS [28–34]. Thus, the WHO [10] has estimated that, currently, in Brazil, an average of 22,000 people lose their lives prematurely annually, owing to exposure to environmental pollutants, particularly in urban agglomerations.

In addition to the various studies that make up the WHO data [10], several authors have demonstrated the benefits of switching from current fuel (diesel oil) to biodiesel in diesel engines. Because in addition to reducing emissions of gaseous pollutants [35], there is a reduction in hospitalization rates for respiratory diseases [36,37]. Moreover, according to the ANP, vehicles that use diesel engines are less than 15% of the total, but correspond to more than 40% of the total gas emissions in Brazil [38].

Thus, this study aimed to present a proposal for reducing emissions, fuel cost, and respiratory disease hospitalizations using environmental cost accounting principles to produce biodiesel from waste cooking oil collected in São Paulo city. A questionnaire was used to survey individuals and restaurant owners, and residual frying oils were collected during the interviews. Data on gaseous emissions, respiratory disease hospitalizations, and their costs were also collected from official health and environmental agencies, and the numbers were correlated to determine how much the hospitalizations and their costs could be reduced via replacing diesel oil with biodiesel.

*1.2. Pathologies Associated with Air Pollution*

The main risk groups that are significantly affected by air pollution and are more susceptible to develop health problems, such as respiratory diseases, are children and the elderly. Moreover, chronic exposure to $PM_{2.5}$ emitted via the burning of fossil fuels, particularly those containing lead, increases the risk of heart and respiratory diseases, and could even cause lung cancer. The high air pollution in large urban centers has also been associated with increased risk factors for cardiovascular diseases, such as cardiac arrhythmias, vasoconstriction and hypertension, myocardial and cerebral ischemia, and the progression of arteriosclerosis [39].

Individuals most susceptible to diseases caused by pollutant emissions are children, the elderly, people with chronic diseases, and those with genetic susceptibility. In addition, pollutants can affect the human fetuses during pregnancy, causing intrauterine growth delays, prematurity, low birth weight, and in the most severe cases, congenital anomalies, and intrauterine or perinatal death [37].

The effects of air pollution on people's health have been researched worldwide, and scholars have attempted to establish the relationship between the photochemical effects of air and the health of the respiratory system and/or allergic conditions. Kunzli et al. [38] reported the increased asthma incidence in adults living in high-traffic regions in Switzerland. Gehring et al. [40] associated the increase in $PM_{2.5}$ levels with the increased asthma incidence in eight-year-old children.

De Marco et al. [25] and Yu et al. [28] reported that $PM_{10}$ and $NO_2$ emissions induced premature deaths, due to cardiovascular and respiratory diseases. Ko et al. [41] associated the increased number of chronic obstructive pulmonary disease (COPD) hospitalizations with increased $SO_2$, $NO_2$, $PM_{2.5}$, and $PM_{10}$ emissions. Andersen et al. [42] and Hu et al. [43] also reported the increase in the number of COPD hospitalizations owing to the increased $NO_2$ emissions and in the emission of pollutants from biomass burning.

Reis et al. [14] noted a very high incidence of lung cancer with $PM_{2.5}$ emissions. Ghering et al. [40] and Pandya et al. [44] reported that high $PM_{2.5}$ levels were associated with the increase in the number of cases of bronchial asthma and allergic respiratory diseases owing to excessive exposure to $PM_{10}$ from burning fuels, particularly diesel oil [44–50]. $PM_{2.5}$ is also associated with the increase in the systolic blood pressure of hypertensive and/or diabetic outdoor workers [20,45–53]. Table 1 summarizes the symptoms and diseases that affect the respiratory tract and that are caused by exposure to gases emitted by burning fossil fuels.

**Table 1.** Gaseous pollutants and their effects on human health.

| WHO Standard | PM$_{10}$ 50 µg/m$^3$ per 24 h or 20 µg/m$^3$ per Year | PM$_{2.5}$ 25 µg/m$^3$ per 24 h or 10 µg/m$^3$ per Year | CO 10 mg/m$^3$ per 8 h | NO$_2$ 200 µg/m$^3$ per 24 h or 40 µg/m$^3$ per Year | SO$_2$ 20 µg/m$^3$ per 24 h | O$_3$ 100 µg/m$^3$ per 8 h |
|---|---|---|---|---|---|---|
| Effects on human health | − enter the nose, throat, trachea, bronchus, and bronchioles; <br> − could cause airway irritation, induce oxidative stress in the lungs, and bronchial tubes, and consequently, could lead to systemic inflammation; <br> − could also cause bronchial remodeling and even cancer after chronic exposure | − all symptoms associated with PM$_{10}$ exposure; <br> − bronchial asthma, allergic respiratory diseases, and increased mortality in children; <br> − increased blood pressure in hypertensive and/or diabetic outdoor workers. <br> − Some authors claim that this is one of the most aggressive pollutants for humans, because these particles could enter the lung alveoli and even the bloodstream | − enters the lung alveoli and bloodstream; <br> − could bind to hemoglobin, which could cause vomiting, nausea, and dizziness; <br> − new-born children are the most susceptible to these problems | − enters the trachea, bronchus, and bronchioles; <br> − is associated with intense vehicle traffic areas; <br> − could cause asthma, rhinitis, and other conditions; <br> − possesses irritating potential and could affect the mucosa of the nose and throat, which could cause coughing and allergies | − affects the upper airways, trachea, bronchi, and bronchioles, and could cause allergic reactions, bronchoconstriction, changes in lung function, and respiratory symptoms; <br> − could cause irritation of the mucosa of the eyes, nose, throat, and respiratory tract; <br> − could cause coughing and increased bronchial reactivity, facilitating bronchoconstriction; <br> − could affect the thyroid | − affects the trachea, bronchus, and bronchioles; <br> − induces respiratory inflammation, airway obstruction, coughing, and discomfort; <br> − has been associated with increased respiratory and cardiovascular problems, particularly in the elderly and children |

Sources: [11,14,21,26,37,42–44].

## 2. Materials and Methods

### 2.1. Questionnaire

In this study, a survey was conducted from 2016 to 2018, and waste cooking oil was collected from the surveyed restaurants and residences in São Paulo city. A number of questionnaires were distributed in different regions of São Paulo city, according to the population and restaurant distributions [7,22,54–56]. Restaurant managers and residents were interviewed using a previously standardized and validated questionnaire. We intended to obtain answers to the following questions:

- Which region of São Paulo city are you located in?
- Do you use frying oil?
- How much frying oil do you use per month?
- What type of frying oil do you use?
- Do you reuse frying oil?
- How many times do you reuse frying oil?
- Are you aware of the environmental impact caused by improper frying oil disposal?
- Are you aware that frying oil could be used to produce biodiesel?
- What is the least expensive type of fuel?
- Would you be willing to store residual frying oil for free collection?
- Would you be willing to store for donate the residual frying oil?
- What type of company would you want to collect your residual frying oil from?

Currently, São Paulo city has 12.18 million inhabitants [57], an average of four individuals per residence (3.045 million residences), and 30,200 restaurants that use oil to prepare meals [58]. A total of 600 and 1500 questionnaires were distributed to restaurant owners and residents, respectively, in different regions of São Paulo city according to the population distribution (%), and the return rate of the questionnaires answered by both residents and restaurant managers exceeded 98%.

Table 2 illustrates the distribution of the questionnaires and the actual population distribution in São Paulo city. Data stratification was similar to the actual stratification of the population in São Paulo city, which ensured that the conclusions drawn would match the real population distribution. The surveys were distributed in a non-random manner to achieve the real stratification percentages; however, some surveys were not returned, and that contributed to the slight differences between the real and actual stratification.

**Table 2.** Distributions of population and survey participants in São Paulo city.

| City Area | Population (Million) | Distribution (%) | Survey Participants (%) |
|-----------|---------------------|------------------|------------------------|
| Eastern | 3.6 | 33 | 32 |
| Southern | 3.1 | 28.5 | 26.3 |
| Western | 1.3 | 11.9 | 12.7 |
| Northern | 2.3 | 21.1 | 23.4 |
| Central | 0.6 | 5.5 | 5.6 |
| Total | 10.9 | 100 | 100 |

Source: SP City Hall [58].

Only one person was interviewed by residence or restaurant, with 2.5% error for residence and 4.0% error for restaurants at a 95% confidence level. The profile of the household interviewees was 58.40% female, 40.85% male, and 0.75% others; of these, 20.20% graduated college, 68.63% graduated high school, 8.87% only graduated elementary school, and the others did not report their education lever or were not literate. Despite the diversity of household respondents, all interviewed restaurant managers graduated college [54–56,59,60].

## 2.2. Biodiesel Production

Silva Filho et al. [15] used a mixture of oil and alcohol (volume ratio of 6:1 or molar ratio of 1:3.2) in a 3.0 L continuous stirred tank reactor. The reactions proceeded at 60 °C using 0.1% NaOH ($m/v$) as the catalyst, while the reaction mixture was constantly stirred for up to 120 min. The amount of formed biodiesel was measured out every 10 min by decanting it into a separating funnel; the light phase was washed with petroleum ether and separated using a Soxhlet extractor.

The following properties of the obtained biodiesel were determined: The specific mass at 20 °C using the ASTM-D4052 method, the flash point using the ASTM-D93 method, the acid value using the Ca 5–40 method, and the moisture content using the Af 2–54 approach. These methods have been described, in detail, in the literature [15,19,36]. The biodiesel yield was calculated from the average molecular weight of oil. The formula used to calculate the biodiesel yield obtained from soybean oil (835 g/mol) and ester (881 g/mol) mixture [15,19] could be written as follows:

$$Yield\ (\%) = \frac{V_{Biodiesel} \times d_{Biodiesel} \times 835}{V_{Oil} \times d_{Oil} \times 835} \times 100 \tag{1}$$

where $V$ and $d$ are the volume and density, respectively, of waste cooking oil and biodiesel.

## 2.3. Data Acquisition on Hospitalization and Gaseous Emissions

The number of respiratory disease hospitalizations and their costs were collected using the TABNET-SUS-online platform. All data were selected the São Paulo city between 2009 and 2018 [29]. The number of monthly respiratory disease hospitalizations in São Paulo city hospitals and their costs were quantified and correlated with the pollutant emissions in the region during that period [7,54,55].

Data on $PM_{2.5}$, $PM_{10}$, $O_3$, $NO_2$, $SO_2$, and CO were collected in the neighborhoods of São Paulo city from 2009 until 2018 in accordance with the National Council of the Environment (CONAMA) standard [61]. Several sensors are placed in various regions of São Paulo city and constantly monitor the air quality, which are sent to a central agency of São Paulo State Environmental Company (CETESB) [62]. The monthly emitted air pollutants that presented the highest impact on residents' health were monitored using the relationships proposed by Arbex et al. [40] and Solé et al. [63], as cited by the WHO [10]. The research was conducted on the Qualar (air quality) page of the CETESB website (http://ar.cetesb.sp.gov.br/padroes-de-quality-do-ar/ (accessed on 5 July 2020), and the monthly pollutant emissions data were filtered according to the metropolitan region (Alto Tietê) [61]. All available data were collected from official sources and represent 100% of the data available on the websites; however, $MP_{2.5}$ monitoring started in 2010 [62]. Data on the rainfall index was collected over the last 30 years [62]. For comparison, the air quality standards presented by the WHO [61] and the air quality index proposed by the CETESB and available on the Qualar page [62] were used.

The following hypothesis was verified: The $PM_{2.5}$, $PM_{10}$, $O_3$, $NO_2$, $SO_2$, and CO emitted by several air sources in São Paulo city contribute to the increased respiratory disease hospitalizations and their costs. Rain is known to decrease air pollution, and therefore, the effect of rainfall was also analyzed to determine whether it was correlated with the decrease in respiratory disease hospitalizations and their costs [40,43,63–66]. If an inverse relationship with the amount of rainfall could be confirmed, then the harmful effects of $PM_{2.5}$, $PM_{10}$, $O_3$, $NO_2$, $SO_2$, and CO should increase in the absence of rainfall [16–18,21]. To verify if there is an effect of each gas parameter on hospitalization, Pearson's correlation was used. The Pearson correlation coefficient was used to perform statistical analysis of the data, and their values were classified as strong, moderate, and weak correlation [10,54,55].

### 2.4. Environmental Cost Accounting Strategies

The cost of biodiesel produced from residual frying oils was determined according to the assumptions of Oliveira Neto et al. [4]; Miranda et al. [19], and Silva Filho et al. [15] as follows:

(a) the population was considered to comprise the number of restaurants (30, 200) and households (3.045 million), according to their respective representative sectors [54,55];

(b) the monthly average amount of waste cooking oil was considered the one reported into questionnaires from surveys for the household and restaurant populations, considering their respective weights;

(c) the monthly average amount of waste cooking oil ($V_{WFO}$) to be collected monthly was calculated using Equation (2):

$$V_{WFO}(\text{L}) = \frac{N_1}{n_1} \times \sum_{j=1}^{n_1} x_j + \frac{N_2}{n_2} \times \sum_{k=1}^{n_2} x_k \tag{2}$$

where $N$ and $n$ are the population and sampling, respectively, and $j$ and $k$ denote households and restaurants, respectively;

(d) the waste cooking oil would be collected by attaching a reservoir to the garbage collection trucks for this specific purpose;

(e) the net value of the logistics planning and waste oil collection cost would be zero;

(f) the waste cooking oil would be donated by the population and restaurants to the City Hall;

(g) conversely, homes and restaurants should not be charged for collecting the waste frying oil, and thus, the net cost of biodiesel for the population and City Hall would also be zero;

(h) the amount of biodiesel obtained would be determined by the monthly volume of collected waste frying oil, multiplied by the yield of the biodiesel production reaction as follows:

$$V_{Biodiesel}(\text{L}) = V_{Oil}(\text{L}) \times Yield \tag{3}$$

(i) the price of traded biodiesel would be used to calculate the profit from its sales as follows:

$$(Profit_{Biodiesel}(\text{L}) = V_{Biodiesel}(\text{L}) \times Price(\text{US\$/L}) \tag{4}$$

(j) biodiesel production would be associated with carbon credits, and 2.5 carbon credits would be attributed per ton of biodiesel. These carbon credits would also contribute to the biodiesel production profits and improve the image of São Paulo city.

This conversion of the energy consumed (or surplus of energy) to carbon credits would be performed using the Official Carbon Credits Calculator developed by the Brazilian GHG Protocol certified by the LRQA Business Assurance [15].

The use of biodiesel would enable the reduction of $CO_2$ emissions and would consequently translate into 2.5 carbon credits for each ton of $CO_2$; the biodiesel carbon credits would be calculated using Equation (5) [15,19]:

$$CC_{Biodiesel}(\text{ton } CO_2) = 2.5 \times m_{Biodiesel} \tag{5}$$

where $m_{Biodiesel}$ is the mass of biodiesel obtained using Equation (6) [12,15].

$$m_{Biodiesel}(\text{ton}) = V_{Biodiesel} \times d_{Biodiesel} \tag{6}$$

where $V_{Biodiesel}$ and $d_{Biodiesel}$ are the volume and density of biodiesel, respectively.

The profit associated with the sale of the carbon credits could be obtained using Equation (5) [15], as follows:

$$CC\ Profit\ (\text{US\$}) = \text{ton}CO_2 \times Current\ Price\ (\text{US\$/ton}CO_2) \tag{7}$$

(k) the profit from the glycerine generated as by-product would also be accounted for. The mass of glycerine ($m_{Glycerine}$) would be calculated from $m_{Biodiesel}$ using its stoichiometric ratio to biodiesel, according to Equation (8), and its profit would be determined using Equation (9):

$$m_{Glycerine}(\text{kg}) = \frac{92}{881} \times m_{Biodiesel}(\text{kg}) \tag{8}$$

and

$$Profit_{Glycerine} = m_{Glycerine}(\text{kg}) \times 5.7 \ (\text{US\$/kg}) \tag{9}$$

(l) the replacement of fossil fuels with biodiesel would reduce the emission of pollutants and respiratory disease hospitalizations by 30% [10]. Thus, it was considered that switching from diesel oil to biodiesel would also reduce the hospitalization-related cost by 30%, and that was considered in the final accounting. Consequently, the decrease in hospitalization cost was calculated as follows:

$$Hospitalisation \ savings \ (\text{US\$}) = 0.30 \times Total \ hospitalisation \ cost \ (\text{US\$}) \tag{10}$$

All fuel prices and hospitalization costs cited in the paper were retrieved from their original sources (ANP, SUS, base year 2019) and were converted to US $ to avoid fluctuations or the effects of inflation during the period following data collection and interpretation. The price of glycerine and carbon credit followed the values traded by the European Union [4,5,15,19].

Subsequently, the environmental cost accounting was summarized, and the data were presented in a table. The data presented the positions and evolutions of São Paulo city as it adopted the ecologically friendly system proposed in this study.

The steps of the eco-friendly process for discarding waste cooking oil are presented below [15,19]:

Stage 1—the current unsustainable stage. At this stage, the environmental effects are associated with the processing of the feedstock and production of waste, and do not include the production costs.

Stage 2—a more sustainable stage in which the company is taking steps to reduce its effect on the environment.

Stage 3—the stage in which operations should not affect the environment.

Stage 4—the stage in which the company is self-sustainable, whereby the environmental accounting balance of its operations generates credits for the company.

## 3. Results and Discussion

### 3.1. Effects of Polluting Gases on Human Health and Healthcare Costs

Figure 1 depicts the annual variations in the $PM_{2.5}$, $PM_{10}$, and $NO_2$ emissions in São Paulo city during the study period. These gases were the ones that were mostly outside the standards established by the WHO, in the period studied. Since the beginning of the environmental monitoring period, the emissions have exceeded the limit set by the WHO. Although the emissions decreased in 2018, the emission values still exceeded the annual WHO-established standards for air quality. Thus, the residents of São Paulo city are more prone to developing respiratory health problems owing to their prolonged exposure to these pollutants, and it is possible that some residents might have acquired chronic diseases or cancer or might have even prematurely died. At the end of 2017, there was a change in the positions of the data collection stations, and this may have influenced the reduction of emissions in the following year.

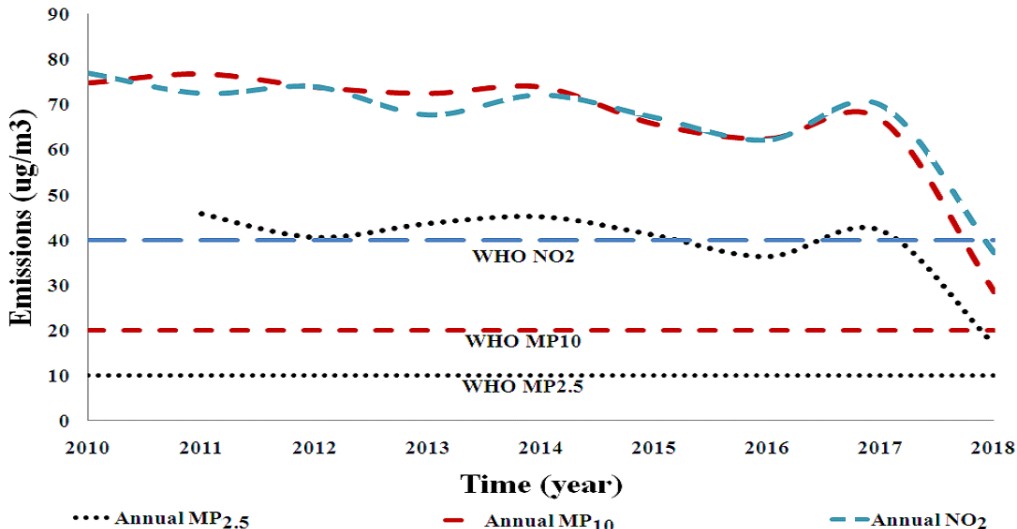

**Figure 1.** Annual PM$_{2.5}$, PM$_{10}$, and NO$_2$ emissions during the study period.

During the study period, $I_{AQ}$ was determined using Equation (2) and the results are presented in Figure 2. The monthly averages, and their lower and upper limits, are represented on the curve. Air quality was poor throughout the study period, even when the recorded levels were very low. Moreover, $I_{AQ}$ was moderately correlated with respiratory disease hospitalizations and their costs. This indicated that all polluting gases affected the number of hospitalizations and their costs [35,59,67,68], and that those exposed to these pollutant gases could develop any of the conditions listed in Table 1 owing to a combined intoxication effect [34,65]. Of all pollutants, PM$_{2.5}$ was responsible for 65.8% of the total number of poor air quality-related hospitalizations, followed by PM$_{10}$ (29.6%) and O$_3$ (4.6%), and these data agreed with the results summarized in Table 3.

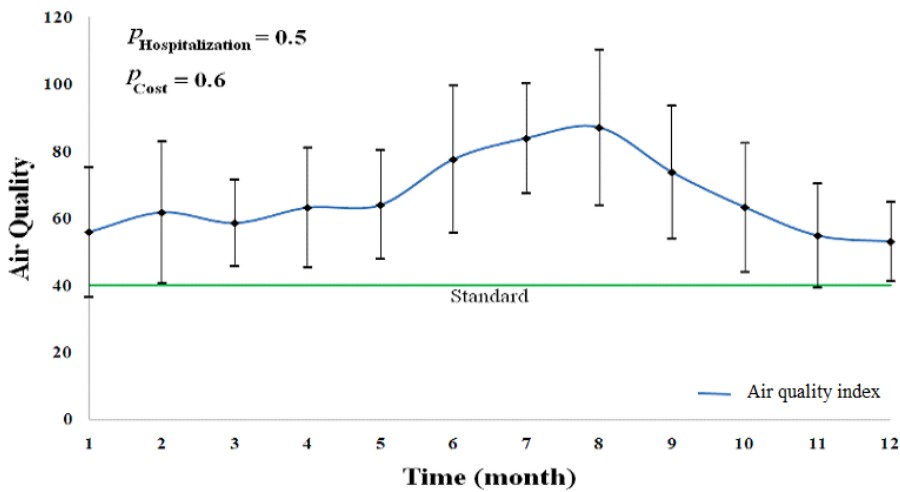

**Figure 2.** Monthly air quality behavior in São Paulo during the period studied.

**Table 3.** Average monthly emissions, hospitalizations, and their costs during the study period (2009–2018).

| Month | World Health Organization Standards | | | | | | Rainfall (mm) | Hospitalizations | Cost (US$) * |
|---|---|---|---|---|---|---|---|---|---|
| | PM$_{10}$ | PM$_{2.5}$ | SO$_2$ | O$_3$ | CO | NO$_2$ | | | |
| | 50 µg/m$^3$ per 24 h | 25 µg/m$^3$ per 24 h | 20 µg/m$^3$ per 24 h | 100 µg/m$^3$ per 8 h | 10 µg/m$^3$ per 8 h | 200 µg/m$^3$ per 24 h | | | |
| January | 59 | 33 | 7 | 94 | 1.2 | 60 | 237 | 1789 | 725,735.73 |
| February | 67 | 37 | 8 | 108 | 1.4 | 71 | 222 | 1846 | 696,734.12 |
| March | 61 | 36 | 9 | 82 | 1.2 | 62 | 161 | 2724 | 901,933.37 |
| April | 64 | 41 | 9 | 81 | 1.3 | 67 | 73 | 3422 | 1,099,880.71 |
| May | 70 | 40 | 10 | 62 | 1.5 | 67 | 71 | 3363 | 1,213,888.22 |
| June | 81 | 51 | 10 | 61 | 2 | 76 | 50 | 3146 | 1,103,469.08 |
| July | 84 | 53 | 11 | 68 | 2 | 86 | 44 | 3150 | 1,063,217.16 |
| August | 94 | 55 | 12 | 87 | 2 | 87 | 40 | 2936 | 1,096,727.15 |
| September | 83 | 48 | 11 | 102 | 1.5 | 76 | 71 | 2835 | 1,057,459.77 |
| October | 74 | 41 | 9 | 103 | 1.2 | 70 | 127 | 2788 | 995,815.47 |
| November | 59 | 35 | 7 | 93 | 1 | 58 | 146 | 2672 | 948,320.17 |
| December | 61 | 36 | 7 | 97 | 1.1 | 63 | 201 | 2264 | 824,124.69 |
| | | | | Correlation coefficient | | | | | |
| Cost | 0.6 | 0.7 | 0.7 | −0.7 | 0.5 | 0.5 | −0.9 | 1 | - |
| Hospitalizations | 0.5 | 0.6 | 0.7 | −0.7 | 0.5 | 0.4 | −0.9 | 1 | |

* 1 USD = RS $3.80.

These pollutants could affect the nose, throat, trachea, bronchus, and bronchioles; they could cause airway irritation and obstruction, coughing, and discomfort, could induce oxidative stress in the lungs and bronchial tubes, and consequently, could lead to systemic inflammation. Moreover, chronic exposure could cause bronchial remodeling and even cancer. In addition, PM$_{2.5}$ could cause bronchial asthma, allergic respiratory diseases, increase mortality in children, and increase blood pressure in hypertensive and/or diabetic outdoor workers. Furthermore, O$_3$ could cause cardiovascular problems, particularly in the elderly and children; however, PM$_{2.5}$ is considered to be the most damaging pollutant to human health [61].

The worst IQA values are observed in August, and this reflects an increase in hospitalizations in the following month, as according to Natali et al. [69], in September there is a peak of hospitalizations for respiratory diseases in São Paulo.

Table 3 summarizes the average monthly emissions, hospitalizations, and their costs during the study period. The emission values represent the monthly averages from 2010 until 2018, and the rainfall values, hospitalizations, and their costs are the monthly averages over 10 years (from 2009 until 2018). No emission data were available before 2009. The overall average monthly number of hospitalizations was 2744, and the average individual hospitalizations cost was approximately US $356.10 per patient. During the study period (2009–2018), 329,329 respiratory disease hospitalizations were recorded in the SUS database, which amounted to US $117.273 million. Both the number of hospitalizations and their costs decreased as the amount of rainfall increased, probably owing to the rainfall improving air quality, as reported by Locosseli et al. [64], Sera et al. [65] and Zhao et al. [66]. According to Duhanyan and Roustan [70], Xu et al. [71] and Zhao et al. [63], the pollutants, gases, and PM are removed from air by below-cloud wet scavenging and precipitating inside the raindrops.

Throughout the study period, the emission rates of PM$_{10}$ and PM$_{2.5}$ were higher than the WHO-established values, when the maximum tolerable must not exceed three times a year. This indicated that the air quality in São Paulo city was poor throughout the study period. Consequently, the population is at risk of developing health problems owing to their exposure to the air pollutants. Thus, the longer the exposure to air pollutants, the greater the damage to human health [17,66]. O$_3$ was another pollutant that exceeded the WHO-set limits on some occasions, while the amounts of the other pollutants were always below the WHO-established limits.

Most emissions were positively correlated with the number of hospitalizations and their costs; moreover, the correlations of the pollutants were either moderate or strong. This demonstrated that such pollutant gases affected the number of respiratory disease hospitalizations, which corroborated the information reported by [17,31,65,66]. The amount of rainfall was inversely proportional to the number of respiratory disease hospitalizations,

which was expected, because rainfall precipitates some pollutants and could improve air quality. However, the amount of $O_3$ was also inversely proportional to the number of respiratory disease hospitalizations, which differed from the data reported by most scholars. Nonetheless, this did not indicate that $O_3$ did not affect air quality, as its concentration exceeded the WHO recommendations [10] for some months of almost every year of the study period.

Therefore, the relationship between the emissions and hospitalizations, that was reported for several cities worldwide, has also been documented for São Paulo city. Thus, according to the WHO [10], reducing the emissions could reduce the respiratory disease hospitalizations costs by at least 30%. If the policy of replacing diesel oil with biodiesel were adopted, São Paulo city could reduce the number of annual respiratory disease hospitalizations by approximately 98,800. Consequently, that could save approximately US $35 million over a 10 years period or US $3,518,191.69 annually.

Although it advocates replacing diesel oil with biodiesel, it must not forget that biodiesel also releases NOx and ultrafine particles, and those could still cause respiratory diseases [28,46,72]. However, biodiesel emissions are less harmful than the diesel oil ones and could reduce the respiratory disease hospitalizations their costs by at least 30% (would be 100% if not polluted); moreover, according to the WHO [10], biodiesel a renewable source [7].

### 3.2. Outcome Drawn from the Survey Data

Figure 3 illustrates the degree of agreement to the questions that addressed the usage, environmental impact, and amount of frying oil used. Figure 4a,b presents the average respondent replies for households and restaurants in São Paulo city, respectively. As noted, 96% and 100% of the households and restaurants interviewed, respectively, reported that they use edible oils to fry foods, and 70% of both categories of interviewees indicated reusing the frying oils.

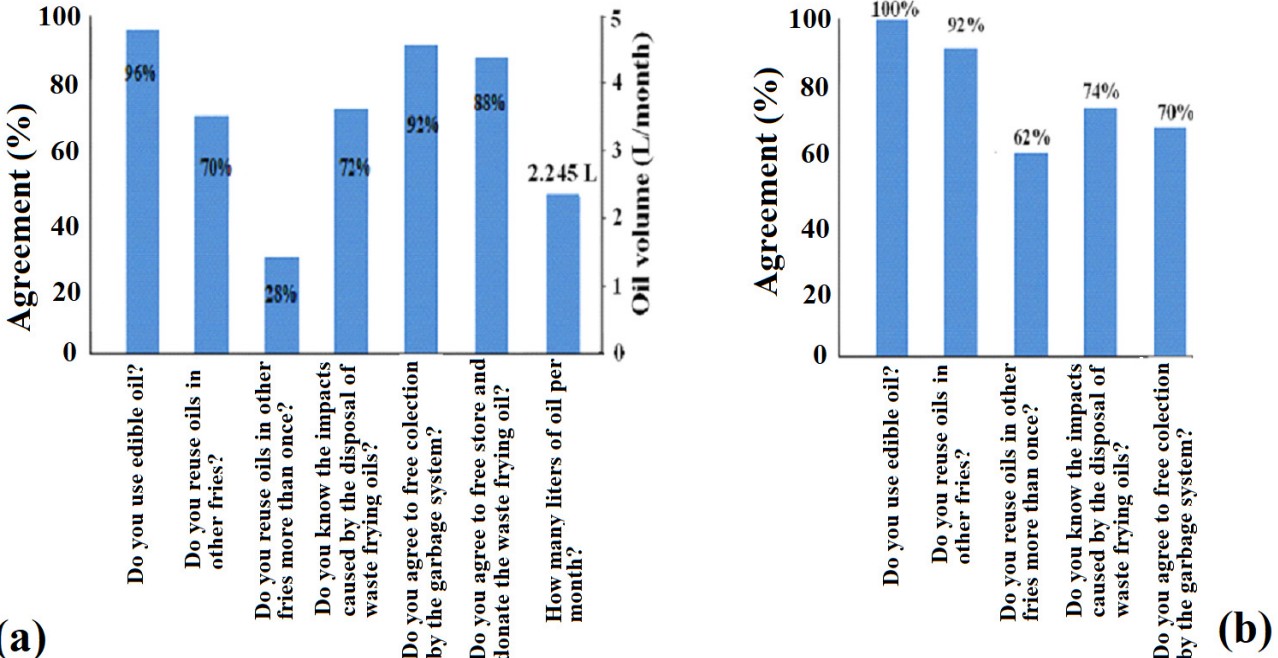

**Figure 3.** Answers to survey questions that addressed the usage, environmental impact, and amount of waste frying oil: (**a**) Households and (**b**) restaurants.

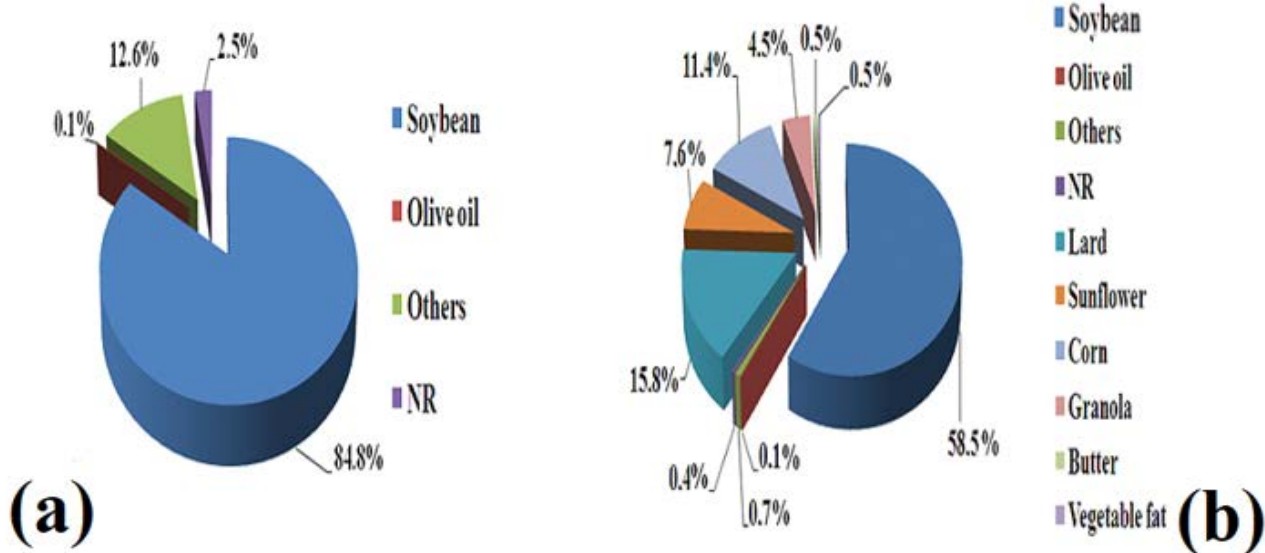

**Figure 4.** Survey data on types of oils used for frying: (**a**) Households and (**b**) restaurants.

For the household interviewees, 70% of all respondents reported reusing the oils, and less than 30% mentioned reusing the same oil at least twice. For the restaurants, 92% of all respondents indicated that they reused the oils, and 62% reused the same oil at least twice. Giraçol et al. [73] reported that the constant reuse of edible oils for various fried foods, exposes the oils to high temperatures for long periods, which causes their oxidation and the release of free radicals, which are subsequently ingested, and consequently, could damage the health of those who consume those foods. Therefore, reusing these edible oils for frying should be avoided.

The percentages of interviewed households and restaurants that were aware of the environmental effects of the irregular disposal of waste frying oils were 72% and 74%, respectively. The most cited environmental effects were: Soil contamination, water pollution, and sewage clogging.

Moreover, the environmental awareness of household respondents was higher than that of the restaurant managers. Approximately 90% of households and only 70% of restaurant managers agreed with the proposal to store and donate for free collect waste frying oils. This demonstrated that the population would be willing to contribute and implement environmentally correct procedures for the disposal and storage of waste frying oils. The average monthly amounts of waste frying oils used in households and restaurants, were 2.245 and 97.663 L, respectively. Therefore, the overall monthly average amount of residual frying oil was determined to be 9794.6 m$^3$, of which 69.89% was contributed by households.

Figure 4a,b presents the percentages of the different types of waste cooking oil used in households and restaurants, respectively, which were compiled after the survey answers were analyzed. Our study indicated that soybean oil was the most commonly used for frying, owing to its low cost, compared to those of other oils, and accounted for 84.8% and 58.4% of the amount of frying oils used in households and restaurants, respectively. The percentage of olive oil used for frying in both households and restaurants was 0.1%, probably because olive oil is typically used as a salad dressing. The types of oils used in restaurants were more diverse and included lard (15.8%), corn oil (11.4%), and sunflower oil (7.6%).

The average composition of the waste cooking oil in São Paulo city was determined using the contributions of the households and restaurants, and the results are presented in Figure 5. Soybean oil accounted for approximately 80% of the entire amount of waste cooking oil, and lard, corn, sunflower, and granola oils accounted for 12%. These results were very useful for studying the reaction kinetics and finding the best conditions to pro-

duce biodiesel from the waste cooking oil collected in São Paulo city. A survey conducted in restaurants in São José do Rio Preto, São Paulo, Brazil, revealed that soybean oil represented 86.2% of the edible oil used for food preparation [74]. Another survey conducted in restaurants located in cities on the coast of São Paulo indicated that soybean oil accounted for 76% of the edible oils used for food preparation, and the other used oils were lard and vegetable fat [75].

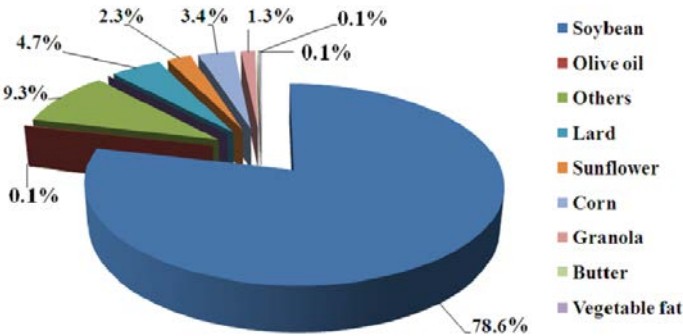

**Figure 5.** The average composition of waste cooking oil in São Paulo city.

Figure 6a,b illustrates the disposal of waste frying oils from households and restaurants, respectively. Only 26% of households had access to selective collection facilities for their waste frying oils (Figure 6a). In addition, 35% of the interviewed restaurant owners stated that they had access to selective collection facilities (Figure 6b). Moreover, significant percentages of the household (52.1%) and restaurant (56%) survey respondents reported discarding waste cooking oil using unregulated methods (pouring it in sinks, sewers, or drains, throwing it away in garbage bags, or disposing of it in yards). However, the percentage of restaurant managers who graduated college was five times higher than the corresponding percentage of household respondents; therefore, it was expected that restaurant managers would be more environmentally aware than the other survey participants. This indicated that environmental awareness was not directly related to the education level of people in São Paulo city. Incorrect disposal of waste frying oils could contaminate rivers, soils, cause significant disruption in the food chain, and affect aquatic life, and therefore, could negatively affect natural resources. Thus, it could be concluded that no public policy currently encourages the correct collection of waste cooking oil in São Paulo city.

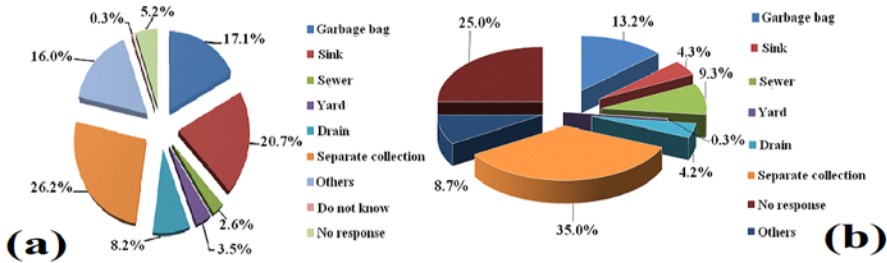

**Figure 6.** Waste cooking oil disposal survey answers: (**a**) Households and (**b**) restaurants.

A study conducted in Goiânia, Brazil, revealed that households discarded 40% of waste frying oils by throwing them away in garbage bags and 20% by pouring them down the sink drain, i.e., 60% of the population irregularly discarded this residue [76]. When interviewing residents in Duque de Caxias, Brazil, it was concluded that 66% of the waste cooking oil was irregularly discarded; indicating an amount similar to that found in São Paulo [77].

Castro et al. [78] determined that 43% of the frying oils used in restaurants in Mogi das Cruzes, Brazil were selectively collected and were used for soap making, which indicated

that the restaurant managers in the area were more environmentally aware than those in São Paulo. However, most waste frying oils were discarded irregularly using the same methods mentioned above, and therefore, caused the same environmental effects as those in São Paulo city. We concluded that Brazilian residents were not aware of any policy on the correct disposal of waste frying oil. Therefore, strategies to increase the environmental awareness, campaigns, and incentives for the selective collection of waste frying oils should be urgently implemented to minimize the damage caused by the incorrect discharge of waste frying oils.

Figure 7 illustrates the perception of the survey participants on the type of company that should collect waste oil. Approximately 57% and 45% of the household and restaurant interviewees, respectively, indicated that they would prefer that the waste cooking oil be collected by public or third-party companies. Moreover, 25–30% of respondents indicated that waste frying oils could be collected by any company, and therefore, expressed no objections to a public or third-party company collecting it. This indicated that approximately 80% of the interviewees would agree that the waste frying oils should be collected by a public company or a partner third party company.

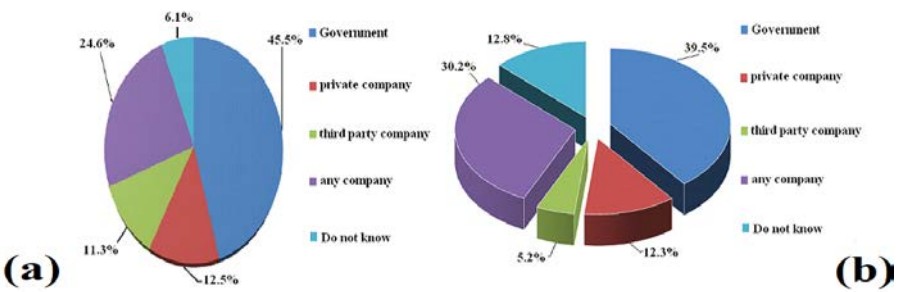

**Figure 7.** Types of companies that interviewees considered that should collect waste frying oils: (**a**) Households and (**b**) restaurants.

A study conducted in Ponta Grossa, Brazil indicated that 98% of the interviewed restaurant managers agreed to the selective waste cooking oil collection by a public company, and 96% of them agreed to donate the waste frying oils [79].

Figure 8a,b depicts the variations in the degree of agreement on the biodiesel questions in the survey between household and restaurant participants, respectively. The survey data revealed that 70% and 62% of the household and restaurant respondents, respectively, believe that producing biodiesel from soybean oil could increase the supermarket price of edible oils. Approximately 66% and 36% of the household and restaurant respondents, respectively, indicated being aware of the possibility of producing biodiesel from waste frying oils. This demonstrated that household respondents were more environmentally aware than restaurant managers. Furthermore, 72% of both the household and restaurant survey participants were aware that biodiesel could be used as fuel in cars, buses, and trucks. When answering the open-ended survey questions, both categories of interviewees indicated that biodiesel could be used as fuel for trucks, buses, bucket trucks, power generators, agricultural and industrial engines. Moreover, both types of respondents believed that biodiesel could replace other fuels, the company responsible for producing biodiesel would be profitable, and that bus tickets could be more affordable if biodiesel were used as fuel for public transportation buses. This demonstrated that both categories of survey participants shared similar ideas about biodiesel.

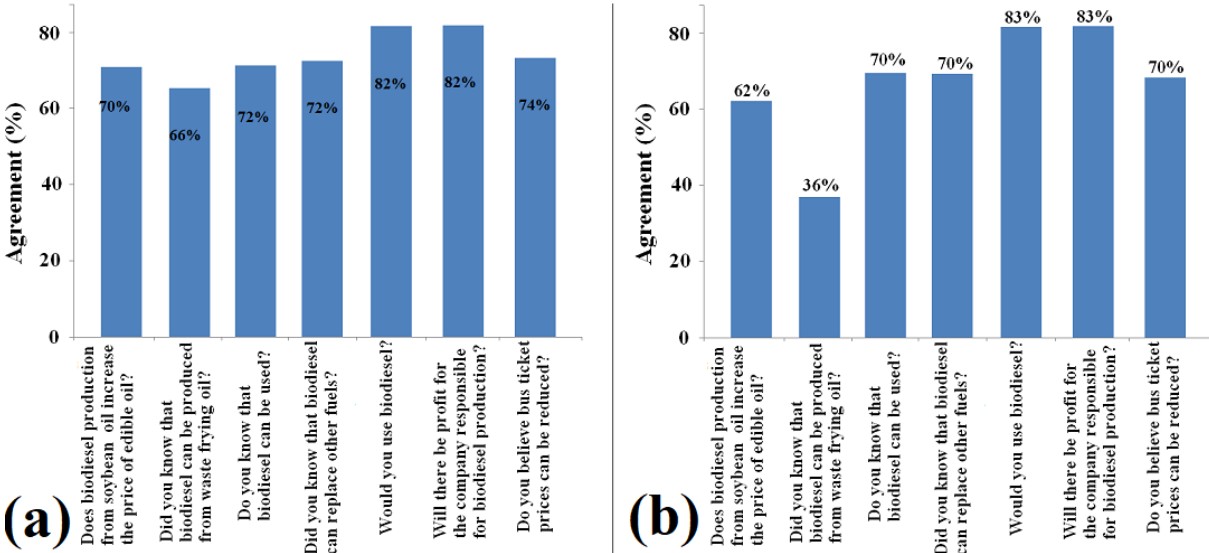

**Figure 8.** Answers to questions on the use of biodiesel: (**a**) Households and (**b**) restaurants.

Thus, it was observed that interviewees intended to act collaboratively towards the storage and collection of residual frying oil, and supported the production of biodiesel mainly by public companies. Therefore, it would be up to the government to elaborate a policy that would inform the population on the correct methods for the disposal and collection of these residues, and to institute awareness campaigns that would promote incentives for the spontaneous and widespread collection of waste frying oils [70].

According to the results presented in Figure 7, a company managed by the São Paulo City Hall would have to collect the waste frying oils. The collection logistics should be the same as that for the collection of garbage, and would simply require the addition of an oil-collection tank to garbage collection trucks, as suggested by Miranda et al. [16] in Campinas and Guarulhos, São Paulo, Brazil. Figure 9 presents photographs of samples collected during the distribution and collection of the surveys that were subsequently used to produce biodiesel. The samples were stored in a laboratory in a temperature range of 15–25 °C. The restaurant waste cooking oil samples were cloudy and included white deposits that comprised flour and fat. The households waste cooking oil samples were cloudy only at low temperatures and clear at 25 °C, and also included small deposits that consisted of flour, meat, and other food residues. The mixture of waste frying oils was cloudy at 15 °C and included a small flour and fat deposit; moreover, at 25 °C the mixture was clear, and its fat and flour deposits were smaller than at 15 °C. Therefore, Figure 10 reveals that oil collection on cold days could be challenging, as the fat and flour residues could hamper the flow, particularly for the waste oils collected from restaurants.

To render this waste oil collection proposal viable, logistic actions are essential, as no specific Federal Law on the disposal of cooking oil exists. Therefore, the proper disposal or recycling of cooking oils by establishments and/or the citizens themselves could be challenging, and therefore, it would be necessary to create concrete conditions that would prevent the discarding of oils into the wild, encourage recycling, and supervise the disposal establishments. Furthermore, tailored vehicles (municipal garbage trucks with collectors) would have to be equipped with appropriate oil collection and transportation containers.

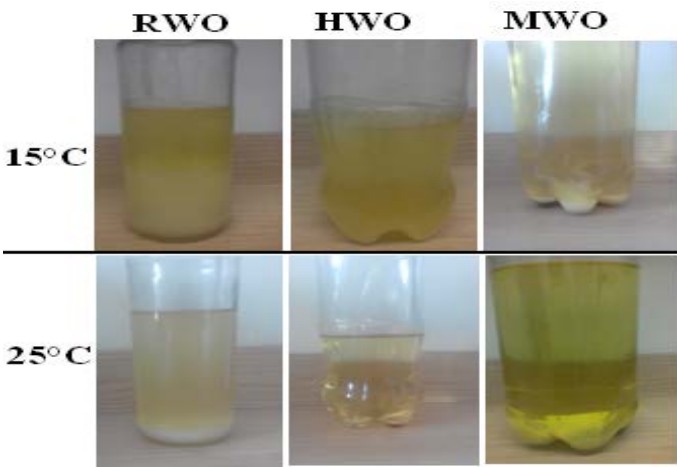

**Figure 9.** Waste cooking oil samples collected from households and restaurants.

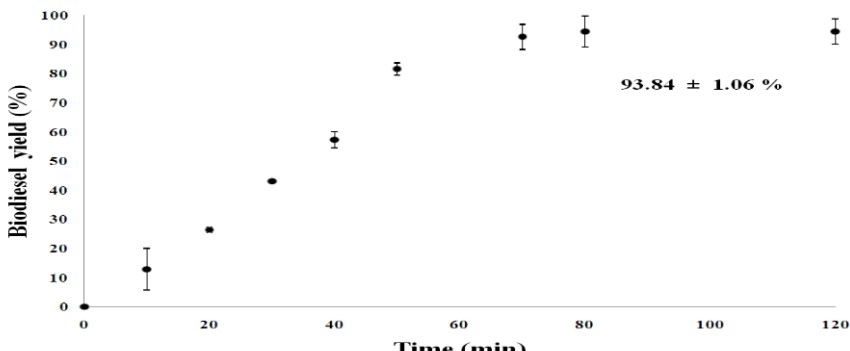

**Figure 10.** Biodiesel formation curve from the restaurant and residential waste cooking oil mixture.

### 3.3. Environmental Cost Accounting Analysis

The total monthly waste cooking oil production of 9794.6 m$^3$ was estimated using the current population of São Paulo city and the average monthly quantities of waste cooking oil reported by restaurants and households (97.663 and 2.245 L, respectively). Table 4 presents the biodiesel quality assessment data based on the mixture of restaurant and households waste oils. All physicochemical properties of the obtained biodiesel were within the National Agency of Petroleum [36] standards. Figure 10 shows the kinetic curve for obtaining biodiesel, which had an average biodiesel yield was 93.84%, and the purity of the obtained biodiesel samples was 98.41 ± 0.28% (the minimum purity should be 96.5%). Thus, 9191.2 m$^3$ (7990.4 t) biodiesel could be produced every month from the total amount of waste cooking oil used collected from restaurants and households in São Paulo city.

**Table 4.** Properties of biodiesel obtained from waste cooking oil mixture.

| Source | Yield (%) | Density (g/mL) | Acid Value (g/100 g) | Moisture (g/100 g) | Flash Point (°C) | Viscosity (mm²/s) |
|---|---|---|---|---|---|---|
| Biodiesel | 93.84 ± 1.06 | 0.8834 ± 0.0251 | 0.3627 ± 0.2517 | 0.023 ± 0.002 | 51.0 ± 0.4 | 4.0 ± 0.5 |
| ANP | - | 0.875–0.900 | <0.8 | <0.5 | >38 | 3.0–6.0 |

According to the São Paulo City Hall [56] data, the city fleet comprises approximately 15,000 diesel vehicles (buses, tractors, trucks, pickup trucks, and others), and it consumes 33,333 m$^3$ diesel oil monthly, which translates into the monthly fuel cost of US $18,392,982.46. This fuel cost was used to evaluate the economic efficiency of the proposal

in this study. Thus, the biodiesel obtained by São Paulo is equivalent to 27.57% of the total needed, and thus, the remaining value for 100% biodiesel will continue to be purchased.

The proposal involves a biodiesel production plant, and a proposed cash flow that includes the acquisition of fuels and production of biodiesel for such a plant is presented in Table 5. It was assumed that the waste cooking oil was donated by the São Paulo residents; therefore, no cost was associated with it. Moreover, the transportation cost was not included because the waste oil was collected at the same time garbage was collected.

**Table 5.** Current monthly balance of capital outflow.

| Description | Quantity (Unit/Month) | Price (US$) | Unit Cost (US$/Month) |
|---|---|---|---|
| Biodiesel plant | 1/120 * | 1,000,000.00 | 8333.33 |
| Maintenance (%) | 4.00 | 8333.33 | 333.33 |
| Oil fuel (L) | 33,333,333.33 | 0.9511 | 31,703,333.34 |
| Ethanol (L) | 1,001,299 | 0.5047 | 505,392.43 |
| | Monthly total cost (US$) | | 32,217,392.44 |
| | Annual total cost (US$) | | 386,608,709.25 |

* is the useful life of the plant = 10 years or 120 months.

For this volume of biodiesel, a biodiesel production plant with an output of 14 m$^3$/h would be sufficient. The market start-up price of a similar plant is US $1 million (Mercado Livre, 2019). Given that the service life of the equipment was 10 years, and the 4% maintenance cost was added to the proposal, the monthly and annual costs of US $8,666.66 and 104,000, respectively, were determined.

Two profit scenarios are presented below, and for both scenarios it was assumed that the oil would be donated:

Scenario 1: The sale of all products generated during the fabrication of biodiesel and its inflow balance is summarized in Table 6.

**Table 6.** Monthly balance of capital inflow for scenario 1.

| Description | Quantity (Unit/Month) | Price (US$) | Profit (US$/Month) |
|---|---|---|---|
| Biodiesel (L) | 9,191,236.31 | 0.7518 | 6,910,358.46 |
| Carbon Credit (t $CO_2$) | 19,976.09 | 9.25 | 184,778.90 |
| Glycerine (kg) | 834,415.72 | 5.70 | 4,756,169.61 |
| | Monthly total profit (USD) | | 11,851,306.93 |
| | Annual total profit (USD) | | 142,215,683.22 |

- The sale of total pure biodiesel could yield a monthly profit of US $6910,358.46.
- The carbon credits and profit were calculated using Equations (5–7), which demonstrated that it would be possible to reduce the monthly and annual $CO_2$ emissions by 19,976.09 and 239,713.08 t, respectively, and to sell biodiesel at the monthly profit of USD 184,778.90.
- Using Equation (8), the monthly and annual amount of produced glycerine was calculated to be 83,415.72 and 10,012,988.64 kg, respectively, which would yield a monthly profit of USD 4,756,169.61, according to Equation (9).
- The total annual and monthly profits would amount to the US $11.851 and 143.652 million, respectively, which would represent a total decrease of 36.79% in the current fuel costs.

Scenario 2: A fraction of the biodiesel produced is used to replace the biodiesel used in the B20 blend, and its out- and inflow balances are summarized in Tables 7 and 8.

**Table 7.** Monthly balance of capital outflow cost for scenario 2.

| Description | Quantity (Unit/Month) | Price (US$) | Unit Cost (US$/Month) |
|---|---|---|---|
| Biodiesel plant | 1/120 * | 1,000,000.00 | 8333.33 |
| Maintenance (%) | 4.00 | 8333.33 | 333.33 |
| Pure diesel (L) | 26,666,666.68 | 0.6897 | 18,392,982.46 |
| Ethanol (L) | 1,001,299 | 0.5047 | 505,392.43 |
| Monthly total cost (US$) | | | 18,907,041.55 |
| Profit for this scenario (US$) | | | 13,310,350.88 |

* is the useful life of the plant = 10 years or 120 months.

**Table 8.** Monthly balance of capital inflow for scenario 2.

| Description | Quantity (Unit/Month) | Price (US$) | Profit (US$/Month) |
|---|---|---|---|
| Biodiesel excess (L) | 2,524,269.64 | 0.7518 | 1,897,971.46 |
| Carbon Credit (t $CO_2$) | 19,976.09 | 9.25 | 184,778.90 |
| Glycerine (kg) | 834,415.72 | 5.70 | 4,756,169.61 |
| Monthly total profit (US$) | | 11,686,504.16 | |
| Monthly total saving (US$) | | 24,996,855.04 | |
| Annual total saving (US$) | | 299,962,260.48 | |

- Pure diesel oil would only be sold by the Brazilian oil company (Petrobras) at 0.6897 US$/L. As the monthly volume of diesel oil would represent 80% of the monthly volume of consumed fuel (26.666 million L), the monthly cost of fuel would be decreased to US $18.393 million.
- Thus, the costs would follow the configuration presented in Table 7, and the difference in fuel outflows and economy with monthly diesel oil purchased of US $13,310,350.88 would be achieved.
- The amount of biodiesel produced would be sufficient to meet the needs of biodiesel (20%) mixed with diesel oil (80%), and an excess of 37.87%pure biodiesel, which would amount to 2,524,269.64 L/month could be sold and generate a monthly profit of US $1,897,971.46.
- Because the profits from the production and sale of glycerine and the carbon credits would be the same, the monthly profit of US $4,776,145.70 should be added.
- The total monthly and annual amounts of US $24.997 million and 300 million, respectively, would translate into the reduction in the total current cost of 78.84%.

The amount that would have to be paid for the biodiesel plant would be insignificant, and it could be recuperated in 1.5–3 days of cash flow. We believe that both scenarios are very good; however, the second one would be twice as economically advantageous as the first. Thus, it would be the most economically viable one. Therefore, the economic and technical feasibility of the production of biodiesel from residual frying oil was demonstrated; moreover, using biodiesel instead of diesel oil would help to reduce pollutant emissions and public spending on health care.

After implementing the policies based on environmental cost accounting, São Paulo could benefit as follows:

- Self-sufficiency in fuel supply;
- Could become a biodiesel supplier;
- Indiscriminate oil disposal would decrease;
- Expenses associated with the cost of the chemicals used for water and sewage treatment would decrease (much of the residual frying oil is currently discarded in inappropriate places that trap the waste into the sewage (Figure 7));
- The emissions of Sulphur and other polluting gases would decrease by more than 98% and 30%, respectively [12,15,73];
- The worldwide criticism associated with the direct use of cooking oil, such as soybean, olive, and corn oil, for the production of biodiesel would be avoided, the prices of these products would no longer increase, and therefore, they would no longer be inaccessible for the neediest communities;
- Agricultural expansion would decrease, as there would be no need to expand the arable area for soybean (and other oilseed) production for biodiesel fabrication;
- The deforestation of forest reserves to increase arable land for the production of oilseeds for biodiesel production would decrease;
- Water consumption for this agricultural expansion would also decrease;
- A non-food product would be obtained, in this case, oils intended for human consumption;
- The annual number of respiratory disease hospitalizations associated with fossil fuel pollutant gases would decrease by more than 9880 hospitalizations;
- Consequently, alleviating the burden on hospitals, and increasing the number of beds available for hospitalization for other diseases; and
- The annual amount of US $3,518,191.69 that would otherwise be allocated to respiratory disease hospitalizations would be saved.

Thus, by mitigating these environmental effects, the image of São Paulo city could be improved.

A summary of the implementation stages of the green cost accounting policy is presented in Table 9, in order from the least to the most sustainable one.

In addition, Natali et al. [68] showed that children and adolescents account for 40% of hospitalizations for respiratory diseases and that these diseases correspond to 30% of total hospitalizations in São Paulo. Thus, as the reduction of gaseous emissions reduces hospitalizations for respiratory diseases, soon, hospital beds will be left to be used for other purposes.

According to Ravina et al. [69], the accounting of the effects of air pollution on human health costs is a useful indicator to support decisions and information at all management levels. Thus, with the economic advantages of ecological cost accounting presented in this work, profits can be redirected towards the expansion or construction of new hospitals, increasing the capacity for hospital beds in São Paulo [7,35,44,67]. This way, it showed how governments can better plan their investments to minimize hospitalizations for respiratory diseases and increase the number of hospital beds available for other diseases, such as COVID-19.

**Table 9.** Summary of the stages of environmental cost accounting presented in this work.

| Field | Stage 1 | Stage 2 | Stage 3 | Stage 4 |
|---|---|---|---|---|
| | Unsustainable | Sustainable | | Fully Sustainable |
| Environmental | — misuse and incorrect disposal of waste frying oils in sewage, water bodies, or dumps; <br> — excessive use of fossil fuel; <br> — emission levels of pollutant gases exceed WHO-acceptable levels | — public awareness via motivational campaigns and incentives for the proper storage and collection of residual frying oils | — SP City Hall should start collecting oils during waste collection, using refuse trucks equipped with frying oil collection reservoirs; <br> — beginning of production and use of biodiesel as fuel; <br> — zero uneven disposal of oil | — continuous operation of biodiesel production plant; <br> — complete replacement of fossil fuel by biodiesel blend; <br> — gain 239,713 carbon credits per year; <br> — reduction in pollutant emissions of at least 30%; <br> — elimination of disposal of frying oil in sewage, garbage, soil, and other unacceptable places |
| Economic | — high costs associated with the fuel purchase, removal of oil during water and wastewater treatment, and hospitalizations of patients owing to conditions caused by the emission of pollutants | — costs associated with advertising, lectures, and building the biodiesel plant | — zero cost associated with the collection of waste frying oil, water, and sewage treatment, and hospitalization sowing to conditions caused by gaseous emissions | — annual gain of US \$299.9 million from the sale of excess biodiesel, carbon credits, and glycerine, and fuel acquisition savings; <br> — decrease in annual hospitalization cost by US \$3.518 million; <br> — estimated total annual profit of US \$300 million |
| Social | — bad reputation owing to increased air, water bodies, and soil pollution; <br> — increased in the numbers of diseased individuals and deaths associated with respiratory diseases caused by exposure to gaseous pollutants <br> — Hospitals with saturated beds by these patients. | — population motivation and driver awareness on the effects of gaseous emissions from fossil fuels | — improved the image of São Paulo city; <br> — slightly decrease the number of hospitalizations and deaths from respiratory diseases caused by exposure to gaseous pollutants; <br> — decrease in the occupation of hospital beds occupied by these patients. | — improve the image of São Paulo city; <br> — boost the self-esteem of residents as they would be contributing to the environmental, social, and economic improvements in the city; <br> — decrease the number of annual hospitalizations and deaths from respiratory diseases caused by exposure to gaseous pollutants by 9880; <br> — Increase in the number of available hospital beds <br> — improvement in the quality of life of the population. |

## 4. Conclusions

When analyzing the database of airborne pollutant gases in São Paulo city, it was noted that the emission levels of $PM_{10}$, $PM_{2.5}$, and $NO_2$ were higher than the WHO-established values throughout the study period, indicating that the air quality in São Paulo city was poor throughout the study period. During the study period, approximately 330,000 respiratory disease hospitalizations were recorded in São Paulo city, which translated into the total cost of US $117 million for the Brazilian government, and positively correlated with the $PM_{10}$, $PM_{2.5}$, $SO_2$, CO, and $NO_2$ pollutant gaseous emissions.

The biodiesel produced from the waste frying oils were within the standard limits, and the total monthly produced volume was estimated to be 9191.2 $m^3$; moreover, the associated annual carbon credits would equal 239,713 t $CO_2$, and the total pollutant emissions would decrease by more than 30%. Environmental cost accounting has revealed that the annual number of respiratory disease hospitalizations could decrease by 9880, and the associated health care cost would decrease by US $3.518 million per year; moreover, the sale of the excess biodiesel, carbon credits, and glycerine, as well as the fuel acquisition savings would result in the annual profit of approximately US $300 million. In addition, the city's reputation and the quality of life of the São Paulo city residents could improve.

**Author Contributions:** Conceptualization, J.C.C.S.; A.C.M. and L.L.H.; Methodology, A.C.M.; L.L.H. and J.C.C.S.; Formal Analysis, L.S.; C.L.K.Y.; A.C.M. and J.C.C.S.; Resources, A.C.M.; L.S.; C.L.K.Y.; F.T.B. and J.C.C.S.; Writing—Original Draft Preparation, D.d.F.C.; F.T.B.; J.C.C.S. and L.L.H.; Writing—Review and Editing; D.d.F.C.; C.L.K.Y.; F.T.B., E.B.T. and J.C.C.S.; Supervision, L.L.H.; F.T.B.; E.B.T. and J.C.C.S.; Project Administration, E.B.T.; F.T.B. and J.C.C.S. All authors have read and agreed to the published version of the manuscript.

**Funding:** This research received the external funding of the National Council for Scientific and Technological Development (CNPq), Brasilia, Brazil, Financial Code: 305987/2018-6.

**Institutional Review Board Statement:** Brazilian ethic review committee—CONEP, Plataforma Brasil: P32432814.8.0000.5511 of 16 April 2014.

**Informed Consent Statement:** Not applicable.

**Data Availability Statement:** Not applicable.

**Acknowledgments:** The authors thank the University of São Paulo (USP) and the Fundação Carlos Vanzolini (FCAV) for financial support. This study was financed in part by the Coordenação de Aperfeiçoamento de Pessoal de Nível Superior-Brasil (CAPES)-Finance Code 001 and National Council for Scientific and Technological Development (CNPq), Brasilia, Brazil, Financial Code: 305987/2018-6. our research group would like to thank, in memory, Silvério Catureba da Silva Filho for his years of contribution to the development and application of technologies for sustainable development. We will never forget you.

**Conflicts of Interest:** The authors declare no conflict of interest.

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
