# Peer review of "Clean Production of Biofuel from Waste Cooking Oil to Reduce Emissions, Fuel Cost, and Respiratory Disease Hospitalizations"

_sustainability, doi:10.3390/su13169185_

Round 1
Reviewer 1 Report
Dear Authors,
I cannot recommend your paper for further consideration due to the significant match (19%) with your earlier paper, ref. [4]. Besides this unfortunate situation, other severe flaws must be improved before considering the resubmission of this work.
- None of you is a medical expert. Consider consulting one before mixing up such topics.
- The paper is too long, discussing too many aspects. It should be either upgraded to a review paper or split into smaller papers with a more clear message with significantly deeper analysis.
- The waste frying oil is a less frequently used term than waste cooking oil - otherwise, the two are identical. Hence, you have missed several relevant literature sources, which already discussed many aspects of this paper. Consequently, the originality is missing here.
- This paper is extremely wordy and requires significant improvement in its English.
Author Response
I cannot recommend your paper for further consideration due to the significant match (19%) with your earlier paper, ref. [4]. Besides this unfortunate situation, other severe flaws must be improved before considering the resubmission of this work.
R: We had modifying or deleting similar texts. This work is the result of JCCSantana's post-doc and is a continuation of [4], which highlighted the relationship between gas emissions with human health
- None of you is a medical expert. Consider consulting one before mixing up such topics.
R: We do not see the lack of researchers in the field of public health as an unworthiness of the article, as the data are made available on the Brazilian public health website and this relationship between emissions and respiratory diseases is already defended by the WHO and several articles (many, even without health researchers). Most authors who publish on the subject are not in the health area, including many of them found in our references.
- The paper is too long, discussing too many aspects. It should be either upgraded to a review paper or split into smaller papers with a more clear message with significantly deeper analysis.
R: We had reduced the text to 11 pages; excluding too many repetitive and conceptual text and figure.
- The waste frying oil is a less frequently used term than waste cooking oil - otherwise, the two are identical. Hence, you have missed several relevant literature sources, which already discussed many aspects of this paper. Consequently, the originality is missing here.
R: we did a search on theScience Direct website and found 23,599 articles to waste cooking oil and 9,709 articles to waste frying oil; so we agreed with you and changed the terms
- This paper is extremely wordy and requires significant improvement in its English.
R: This text had already been corrected by a native speaking English via Elsevier Language Editing Services. The title was changed at the editor's request to suit the theme of the special edition
We thank you for your suggestions as they will improve the quality of our article.
Reviewer 2 Report
This manuscript is obviously not ready for review, first, it does not have page numbers; second, careless typos are found, e.g “cos” in the abstract, “MP”s in Figure 2; third, the manuscript is unnecessarily long and contains too much textbook like fundamental information (e.g. s.1 and subsections); fourth, substantial point form writing is found towards the end of s.3.2. However, even if such carelessness is rectified, major problems are found with the reliability and accuracy of the source of some important variables.
Regarding the questionnaire surveys, there is no information (other than the sample was stratified) on how the respondents were selected, has ethical approval been obtained, what kind of “standardization” and “validation” were performed and performed by who? Who in a restaurant and who in a household were invited to supply the information? The questions were also very unskillfully drafted. Take for example the question “what is the least expensive type of fuel?” This is a vague question as the usage of the fuel is not specified. Therefore whatever answers were obtained, they would not be accurate. Another example is “Would you be willing to donate the residual frying oil?”. However, the results of this question from both sectors are not reported in the manuscript. To support the feasibility of used cooking oil donation, the authors simply cite from a 2017 publication for a high 96% of willingness to donate. Why are the data from the survey cited instead? Because the data from their own survey do not lend support to used cooking oil donation? It appears that the authors have not fully disclose the entire questionnaire in the manuscript. A number of variables mentioned in Figure 9 were not mentioned in section 2.1. Intentional or not, the authors have misled the respondents by not providing an exhaustive list of possible responses to many of the questions. Take the example of “Would you use biodiesel?”. The authors only present the “agreement” % (Figure 9). However, are the agreements conditional or unconditional? It appears to me that the authors are trying to paint a picture of unconditional full support from the residents and restaurants to donate used cooking oil and to use biodiesel to make their study well-justified.
A major question that I have is why the authors research on “frying oil” and not “cooking oil”. What differences are there between the two? Even if it is the intention of the authors to study frying and not cooking oil, I do expect an explanation from the authors. However, there is no such explanation in the manuscript. More important, since in the questionnaire, a lot of questions on frying oil were asked, were all the respondents understand “frying oil” in the same way as the researchers? In Asia, waste cooking oil has a market and there are companies specializing in buying waste cooking oil from restaurants.
What quality assurance procedures were used in the survey? How can an individual household or even a restaurant owners be able to estimate accurately the quantities (in liters) of waste cooking oil that they will produce? Were the reported quantities verified?
The environmental accounting procedures were also not clearly explained. For instance, what are xj and xk in Eq.(3)? Why were transport and labour costs of waste oil collection not included in the calculation? It also appears to me that the authors assume that 100% of the reported quantities of waste oil is collectable. How can this be realistic?
Author Response
This manuscript is obviously not ready for review, first, it does not have page numbers; second, careless typos are found, e.g “cos” in the abstract, “MP”s in Figure 2; third, the manuscript is unnecessarily long and contains too much textbook like fundamental information (e.g. s.1 and subsections); fourth, substantial point form writing is found towards the end of s.3.2. However, even if such carelessness is rectified, major problems are found with the reliability and accuracy of the source of some important variables.
R: The source is government websites, all accredited and using the standards of the national environmental council, based on the Environmental Protection Agency, EPA
Regarding the questionnaire surveys, there is no information (other than the sample was stratified) on how the respondents were selected, has ethical approval been obtained, what kind of “standardization” and “validation” were performed and performed by who? Who in a restaurant and who in a household were invited to supply the information? The questions were also very unskillfully drafted. Take for example the question “what is the least expensive type of fuel?” This is a vague question as the usage of the fuel is not specified. Therefore whatever answers were obtained, they would not be accurate. Another example is “Would you be willing to donate the residual frying oil?”. However, the results of this question from both sectors are not reported in the manuscript. To support the feasibility of used cooking oil donation, the authors simply cite from a 2017 publication for a high 96% of willingness to donate. Why are the data from the survey cited instead? Because the data from their own survey do not lend support to used cooking oil donation? It appears that the authors have not fully disclose the entire questionnaire in the manuscript. A number of variables mentioned in Figure 9 were not mentioned in section 2.1. Intentional or not, the authors have misled the respondents by not providing an exhaustive list of possible responses to many of the questions. Take the example of “Would you use biodiesel?”. The authors only present the “agreement” % (Figure 9). However, are the agreements conditional or unconditional? It appears to me that the authors are trying to paint a picture of unconditional full support from the residents and restaurants to donate used cooking oil and to use biodiesel to make their study well-justified.
R: Table 2 corresponds exactly to the stratification of the samples, showing the percentages of the population and the distribution of questionnaires in the strata. The questionnaire was validated in 2011 by Giraçol et al (2011) and also used by Silveira et al. (2018). The text of the questionnaire has been abbreviated as it is longer than 4 pages. There are also issues of income, gender, educational background, but we avoid inserting for ethical reasons. The question about the price of fuel was considered important to keep in the original questionnaire, as there are other questions that address the donation of frying oil to the city hall. We think that this question could influence the donation, because if people know that there is a good financial gain, they might want to not donate the oil. The questions presented in these figures were elaborated on a Likert scale.
You're right, it doesn't appear in the picture or in the text that it's a free collection. The questions specify free collection and storage; two separate questions addressed the free collection and storage of waste; it addressed to store and collect free of charge for the city of São Paulo's garbage collection system. We had modifying the text of the figures and article for a better understanding. So, it has been changed to “you agree to free collection by the garbage system” and “store and donate...” needed to be improved to be clear, in Figure 3. We have the questionnaires in Portuguese to verify all respondents' answers
A major question that I have is why the authors research on “frying oil” and not “cooking oil”. What differences are there between the two? Even if it is the intention of the authors to study frying and not cooking oil, I do expect an explanation from the authors. However, there is no such explanation in the manuscript. More important, since in the questionnaire, a lot of questions on frying oil were asked, were all the respondents understand “frying oil” in the same way as the researchers? In Asia, waste cooking oil has a market and there are companies specializing in buying waste cooking oil from restaurants.
R: The two are accepted as being the same thing, when accompanied by the word waste. We had changed it to waste cooking oil at the request of another reviewer. In Brazil there are no companies specialized in the use of used frying oil. We want to provide a sustainable destination for this waste, as biodiesel is obtained from cooking oil, which makes it expensive because it competes with food
What quality assurance procedures were used in the survey? How can an individual household or even a restaurant owners be able to estimate accurately the quantities (in liters) of waste cooking oil that they will produce? Were the reported quantities verified?
R: We had this text inserted after table 2 “Only one person was interviewed by residence or restaurant, with 2.5% error for residence and 4.0% error for restaurants at a 95% confidence level.”
The environmental accounting procedures were also not clearly explained. For instance, what are xj and xk in Eq.(3)? Why were transport and labour costs of waste oil collection not included in the calculation? It also appears to me that the authors assume that 100% of the reported quantities of waste oil is collectable. How can this be realistic?
R: In the formula, the sub-indices only refer to the quantities of oils coming from homes or restaurants. In Brazil, garbage collection is free and is carried out by the mayor who pays the salaries and other associated costs... a tank will be inserted to collect the waste, so there is no cost associated with collecting the waste frying oil. This was already written in the item d) the waste cooking oil would be collected by attaching a reservoir to the garbage collection trucks for this specific purpose
We thank you for your suggestions as they will improve the quality of our article.
Reviewer 3 Report
The work raises a field of action for the manufacture of biodiesel and improvement of air quality in Brazil, and therefore the reduction of respiratory diseases. It is a very interesting topic, although it does not echo the problems that have been detected in other cities in more developed countries when collecting this waste.
In addition, the work is difficult to read, it may be too long. Especially the introduction should be reduced, being more concise and limiting its study to the objectives of the work. Similarly, it would be essential to reduce the text of the results, eliminating photos that do not contribute much and reworking the graphics that look distorted and mislead the reader.
Proof of the extent of the work and the possibility of reduction is that the conclusions are scarce.
The main weakness of the work is wanting to study two aspects in the same study, on the one hand the possibility of using frying oil as biodiesel and on the other hand, assessing when it would save this in respiratory diseases in Brazil, in my humble opinion there are two aspects with enough entity to analyze them separately.
Author Response
The work raises a field of action for the manufacture of biodiesel and improvement of air quality in Brazil, and therefore the reduction of respiratory diseases. It is a very interesting topic, although it does not echo the problems that have been detected in other cities in more developed countries when collecting this waste.
In addition, the work is difficult to read, it may be too long. Especially the introduction should be reduced, being more concise and limiting its study to the objectives of the work. Similarly, it would be essential to reduce the text of the results, eliminating photos that do not contribute much and reworking the graphics that look distorted and mislead the reader.
Proof of the extent of the work and the possibility of reduction is that the conclusions are scarce.
The main weakness of the work is wanting to study two aspects in the same study, on the one hand the possibility of using frying oil as biodiesel and on the other hand, assessing when it would save this in respiratory diseases in Brazil, in my humble opinion there are two aspects with enough entity to analyze them separately.
R: We had reduced the text to 11 pages; excluding too many repetitive and conceptual text, including the conclusion. One of the graphs has been eliminated, as its data are in the following table. These analyzes have already been done separately in our articles [4, 12], now we propose the mix based on the association that the WHO makes between pollutant emissions and respiratory diseases. Brazil is currently using only 10% biodiesel blended with diesel oil and our research group wants to continue insisting that the use of B100 is environmentally, economically and socially viable, so that governments make this switch to biofuel as soon as possible.
We thank you for your suggestions as they will improve the quality of our article.
Reviewer 4 Report
My main concern is the authors focus on health but the volume of cooking oil would (if my rough estimates are correct) provide only 1/2 of 1 day per month of substitute for diesel oil in the metro city. This is probably not enough to make a major impact on hospital costs or health, though it would offer mild mitigation. It is still a good idea if profitable as it would reduce pollution from cooking oil disposal. See my lengthier comments which support a high health and mortality cost from fossil air pollution.
Author Response
My main concern is the authors focus on health but the volume of cooking oil would (if my rough estimates are correct) provide only 1/2 of 1 day per month of substitute for diesel oil in the metro city. This is probably not enough to make a major impact on hospital costs or health, though it would offer mild mitigation. It is still a good idea if profitable as it would reduce pollution from cooking oil disposal. See my lengthier comments which support a high health and mortality cost from fossil air pollution.
R: São Paulo City Hall fleet consumes 33.333 million L diesel oil monthly and 9,191.2 m3 /month (9,191,200 L/month) of biodiesel could be produced or 27% of the fuel required monthly. in Brazil we only use 10% biodiesel mixed with diesel oil, however the proposal foresees the total replacement of diesel oil by biodiesel. But a gain for the city hall through the reuse of waste frying oil to obtain biodiesel, the other 73% will continue to be purchased. And several other expenses with fuels can be deducted from the gains with: the reduction of costs with the acquisition (or sale) of biodiesel; the acquisition of carbon credits; the sale of glycerin and; the possibility of reducing the cost of hospitalization of patients for respiratory diseases.
We thank you for your suggestions as they will improve the quality of our article.
Round 2
Reviewer 1 Report
Dear Authors,
Since you have committed plagiarism, it is highly unethical to re-evaluate your work until you score a 'pass.' Moreover, the first three out of my four comments were superficially resolved. Hence, my recommendation is still 'reject.'
Author Response
Sorry for disagreeing, but I didn't commit plagiarism, three paragraphs had some similar words, but they weren't quite the same. Just compare the articles, which are mine. I deleted 7 pages because you and another reviewer thought the text was too big. The figures repeated what was in the tables, so it didn't make sense to keep them. Thank you for your comments
Reviewer 3 Report
The paper has been improved. The reviewer considers that the paper could be publish
Author Response
Thank you very much!
Reviewer 4 Report
The only minor request (not condition to proceed) is that the astonishing 50% reduction in 2018 levels of pollution be explained - if possible - in a footnote. If this was due to a stricter diesel fuel standard, unusual rainfall, or a change in fuel use by urban vehicles and buses, it should be noted. If none of these, it is still so unusual after years of relative stability that some comment would be helpful.
Author Response
In the years 2016 and 2017 there was a lot of criticism from the Brazilian media, after the WHO reports publicized the very bad situation of air quality in the city of São Paulo. As a solution, in the following years the government of the state of São Paulo changed the locations of the online data collection stations and, as noted, artificially changed the values. Then, those of us who work with these data, we noticed this "fraud", but the Brazilian media hasn't noticed it yet. We avoid making derogatory comments so that the government does not bar our access to online data. So, we don't want to make that comment in our text. I ask you to forgive us.